# Implications of Antigen Selection on T Cell-Based Immunotherapy

**DOI:** 10.3390/ph14100993

**Published:** 2021-09-29

**Authors:** Faye A. Camp, Jill E. Slansky

**Affiliations:** Department of Immunology and Microbiology, University of Colorado School of Medicine, Aurora, CO 80045, USA; Faye.Camp@CUAnschutz.edu

**Keywords:** neojunctions, HERV antigens, tumor-specific antigens, tumor-associated antigens, single nucleotide variants, alternative splicing-derived neoepitopes, T cell-based immunotherapies

## Abstract

Many immunotherapies rely on CD8+ effector T cells to recognize and kill cognate tumor cells. These T cell-based immunotherapies include adoptive cell therapy, such as CAR T cells or transgenic TCR T cells, and anti-cancer vaccines which expand endogenous T cell populations. Tumor mutation burden and the choice of antigen are among the most important aspects of T cell-based immunotherapies. Here, we highlight various classes of cancer antigens, including self, neojunction-derived, human endogenous retrovirus (HERV)-derived, and somatic nucleotide variant (SNV)-derived antigens, and consider their utility in T cell-based immunotherapies. We further discuss the respective anti-tumor/anti-self-properties that influence both the degree of immunotolerance and potential off-target effects associated with each antigen class.

## 1. Introduction

The current pillars of cancer treatment include surgery, chemotherapy, radiation therapy, targeted therapy, and more recently, immunotherapy. The emergence of immunotherapy as a fifth pillar in cancer treatment is the result of research spanning decades that have advanced our understanding of T cell-mediated immunity, some of which will be touched upon in this review. This, in conjunction with advances in T cell-engineering, genomic sequencing, and computational biology have made it possible to begin to translate these discoveries into meaningful and efficacious clinical outcomes. The T cell-mediated immunotherapies used today include adoptive cell therapy (ACT), anti-cancer vaccines, and immune checkpoint inhibitor (ICI) therapy. Paramount to both the efficacy and potential toxicity of T cell-based immunotherapies are the antigens they target. This review will discuss the various types of cancer antigens, focusing on antigens from neojunctions and retroelements in the genome, and the implications they have on the outcome of T cell-based immunotherapy.

## 2. T Cell-Mediated Immunity

As T cells develop in the thymus, they undergo positive and negative selections that center on the strength of the interaction between the thymocytes’ T cell receptor (TCR) and the peptide:MHC presented by antigen-presenting cells (APCs). Briefly, high-affinity interactions result in negative selection by apoptosis or redirection to the Treg lineage, generating central tolerance against those antigens. Conversely, weak interactions are necessary for positive selection, whereas no interaction results in death by neglect. This intricate process ensures an organism has a repertoire of T cells that can recognize unique peptide:MHC complexes on cells, but not complexes with self-derived peptides, which may lead to autoimmune pathologies [1]. The positively-selected T cells, equipped with their unique TCR, surveilled the body for their cognate peptide:MHC to defend against infection and neoplasia.

Paradoxically, as the immune system eliminates neoplastic cells, it also acts as a selection force fostering immune escape [2,3]. This process of immunoediting can result in uninhibited tumor cell growth and clinically detectable cancer. Therefore, the focus of many immunotherapies has been on reactivation of exhausted CD8+ T cells, antigen discovery, and engineering of antigen-specific T cells.

### 2.1. Cancer Antigens

One of the first indications that T cells have the potential to treat cancer came from the clinical observation that some tumors regress in melanoma patients after infusions of adoptively transferred autologous cytotoxic T cells (CTLs) with IL-2 [4,5]. This fueled the identification of the tumor antigens responsible for this outcome. Since then, many cancer antigens have been identified that can be broadly divided into two overarching categories: tumor-associated antigens (TAAs) or tumor-specific antigens (TSAs).

TAAs are self-antigens expressed differentially by cancer cells and include antigens that are transcriptionally silent in adults such as cancer germline-testes antigens (e.g., NY-ESO-1 in up to 40% of breast cancers [6]), cell line differentiation antigens (e.g., MART-1, in up to 90% of melanomas [7]), antigens that are simply overexpressed or amplified in cancer cells compared to normal tissue (e.g., amplification of Her2/neu in 15–30% of breast cancers results in up to two million Her2 receptors expressed on the tumor cell surface [8]) as well as tumor-associated carbohydrate antigens (TACAs) [9,10]. TAA-directed immunotherapies face two fundamental challenges: (1) the activation of T cells against self-antigen and (2) on-target, off-tumor toxicities. Moreover, T cells specific to TAAs may be present in low numbers or may have simply been deleted from the repertoire during development in the thymus. Therefore, the use of strong adjuvants and co-stimulation is often necessary which can have significant, even catastrophic, on-target, off-tumor toxicities ranging from vitiligo [11] to death [12]. It is these challenges and safety concerns that make TSAs a more desirable immunotherapy target.

TSAs are expressed uniquely by the tumor and therefore are “foreign” to the immune system. These novel epitopes are introduced into the immunopeptidome from either virally encoded proteins (such as oncoproteins and human endogenous retroviruses, HERVs) or from accumulated mutations within the cancer cells genome (neoantigens) [13]. Neoantigens have classically been defined as non-synonymous single-nucleotide variant (nsSNV)-derived peptides that can expand T cell populations when presented on MHC class I (and MHC class II). Although the majority of neoantigens are patient-specific, they may also be shared between patients. Shared neoantigens are often the result of a mutation in a strong driver oncogene, as seen in the KRAS variants, G12D and G12V, which account for up to 60–70% of pancreatic cancers and 20–30% of colorectal cancers (CRCs) [14]. Tumor-specific antigens are advantageous because they have no associated immunotolerance against them as they have not undergone a thymic selection and are therefore more likely to elicit a strong tumor-specific CTL response. Further evidence supporting the utility of tumor-specific neoantigens as targets of T cell-mediated immunity comes from studies involving immune checkpoint inhibitor (ICI) therapy. A clear correlation between a favorable therapeutic outcome of ICIs, such as anti-CTLA4 and anti-PD1, and mutational burden have been identified in several cancer types, including non-small cell lung cancer (NSCLC), melanoma, and bladder cancer [15]. Despite the clear rationale for targeting neoantigens therapeutically, there are still limitations associated with these antigens as elaborated on below.

### 2.2. Identification of Single-Nucleotide Variant-Derived Neoantigens

Targeting cancer cells through neoantigen-T cell interactions has been appreciated for some time but neoantigen identification has been hindered by technological challenges. Recent advances in the genomic sequencing and computational biology sectors have profoundly impacted our ability to discover neoantigens. Prior to these advances, TAAs and neoantigens were identified through laborious and time-consuming screening of target cells that had been transfected with autologous HLA and cDNA libraries with patient-derived tumor-infiltrating lymphocytes (TILs) [5]. Today’s bioinformatic methods utilize sequencing and computational prediction models to identify neoantigens [16]. As elaborated on below, these methods: (1) identify non-synonymous somatic variants, such as single nucleotide variants (SNVs) and insertions and deletions (indels), through the alignment of whole-exome sequencing (WES) from tumor and germline DNA, (2) quantify the relative expression of somatic variants via RNAseq analysis of the tumor sample, (3) predict peptide:MHC binding computationally and/or discover peptides empirically through LC-MS/MS tumor immunopeptidomics, (4) interrogate the immunogenicity of putative neoantigens through CTL screening assays [17,18,19,20] (Figure 1).

In silico MHC class I, prediction tools are widely used in the identification of neoantigens (Table 1). These tools generate a list of peptides from known protein sequences that bind a particular MHC allele using artificial neural networks (ANNs) that are trained on binding affinity data, eluted peptides from mass spectrometry (MS), or both [21]. The affinity with which a peptide binds to an MHC molecule is one determinant of its immunogenicity. In general, high-affinity peptide:MHC interactions are associated with more robust immune responses. Although there are exceptions, previous studies have indicated a binding affinity threshold of 500 nM (or higher) is needed to be considered as a potential immunogenic peptide [22]. Therefore, ANNs trained on binding affinity for MHC class I scan amino acid sequences for 8 to 11-mers that fit in the peptide-binding groove and predict the strength of the interaction with that specific MHC allele. ANNs that are trained on eluted peptides from MS incorporate additional information in the prediction algorithms such as peptide-processing steps through the proteosome and the length of naturally presented peptides [23]. The major limitation surrounding MHC-binding prediction tools is their inability to incorporate the relative abundance of peptide:MHC expression at the cell surface as well as accurately predict immunogenicity as determined by T cell interactions. Therefore, depending on the mutational burden of the sample, this method alone has the potential of generating numerous false positives, the screening of which may become cost-prohibitive or be hindered by the availability of autologous T cells.

An alternative, or supplemental approach, involves eluting peptides directly from the MHC molecules on a tumor sample and interrogating the naturally presented mutated peptides via MS. The major limitation surrounding this approach is sensitivity and an inherent bias towards detecting the most abundant peptides [31]. Consequently, if a peptide is derived from a low abundant transcript or from a subclonal population, there is a high potential of it going undetected. This limitation is not trivial given that a TCR can react to as little as a single (or few) peptide:MHC complex [32].

Neither in silico prediction algorithms nor MS are capable of identifying the epitopes that are both recognized by T cells and elicit a CTL response. Therefore, putatively identified neoantigens must undergo further empirical screening to determine their utility in a T cell-based immunotherapy. These screening assays display antigens using different methods such as: (1) pulsing of target cells with peptide (2) tandem minigenes and (3) peptide-loaded multimers. The simplest assay involves pulsing autologous APCs or HLA-matched target cells with individual or pooled peptides. Subsequently, TILs are screened for markers of recognition and cytotoxicity such as proliferation via flow cytometry, target cell killing via LDH assays, and IFN*γ* production via flow cytometry or ELISpot. An alternative approach, developed by Lu et al. [33] does not rely on in silico predictions. Rather, each mutation identified from the WES alignment along with the flanking 12 amino acids are strung together to generate a tandem minigene. These tandem minigenes are transfected into autologous APCs and co-cultured with patient-derived TILs and screened for cytotoxicity. This method uniquely mimics natural antigen presentation on both MHC class I and II. A final approach utilizes mutant peptide-loaded tetramers to identify antigen-specific TIL; these assays provide no direct functional information about the responding T cells, however.

Studies in a variety of cancer types, including melanoma, renal cell carcinoma, squamous cell carcinoma, and NSCLC, have demonstrated the correlation between mutational burden, immunogenicity, and positive response to ICI therapy. These studies underscore the degree of T cell immunosuppression within the tumor microenvironment and the potential of endogenous T cells to recognize neoantigens upon reactivation [15,34]. The relationship between these factors in breast cancer, however, has remained inconclusive. Thus, Morisaki et. al. profiled 31 breast cancer samples for somatic mutations and neoantigen expression using the pipeline described above [35]. The number of nsSNVs varied between patients, but overall, 1976 nsSNVs were detected across all 31 samples, accounting for 62.4% of the total exonic mutations. Furthermore, the top three genes containing nsSNVs were TP53 (13 samples), PIK3CA (5 samples), and TTN (5 samples). They also identified five recurrent nsSNVs between at least two samples in the following genes: AKT1, ARAP3, NITCH3, PIK3CA, and SLC35E2. From these nsSNVs, they went on to computationally predict putative neopeptides which also varied in number between patients but correlated with a mutational burden. Interestingly, triple-negative breast cancer (TNBC) samples had significantly more predicted neoepitopes (*n* = 13; median = 150, range: 9–440) compared to non-TNBC (*n* = 18; median = 51, range: 6–196) (*p* < 0.05). Lastly, the immunogenicity of 10 TNBC-derived putative neoepitopes was interrogated by coculturing patient-derived peptide-pulsed DCs with autologous lymphocytes. Two of the 10 selected peptides elicited significantly higher cytotoxicity than wildtype peptides (*p* < 0.01) and significantly higher IFN*γ* expression against autologous target cells than the wildtype control (*p* < 0.05) [35]. This study highlights two important aspects of neoantigen identification (1) peptide:MHC binding algorithms produce high numbers of false positives and, (2) the identification of a true neoantigen lies in the cytotoxic T cell response.

### 2.3. Neoantigens Derived from Alternative Splicing

Alternative splicing generates mature mRNA transcripts that encode genes with divergent functions. This process is directed by well-conserved motif sequences at splice sites located at the intron-exon borders. The patterns of alternative splicing include exon skip, alternative 5′ or 3′ splice site, intron retention, and mutually exclusive exons (Figure 2) [36,37]. Mutations in either the motif sequences (cis-elements) or in the splicing factors that associate with them (trans-elements) can result in profound changes in splicing patterns within a cell. Recent analyses of the human transcriptome have shown that alternative splicing is increased in cancer samples compared to matched normal samples by up to 30% [38]. Subsequent analyses have identified a number of mutations in cis- and trans-acting elements that form recurrent alternatively spliced isoforms within and between cancer types. These splicing events generate functional mRNA (that is translated to protein) or mRNA that is targeted for degradation, both of which have direct roles in promoting tumorigenesis and cancer cell survival [39,40]. For example, mutations in the splicing factor, SF3B1, cause the use of an alternative 3′ splicing site, typically 30 nucleotides upstream of the wildtype splicing site of a tumor suppressor gene, BRD9. The new open reading frame introduces a stop codon in BRD9 and leads to mRNA degradation through nonsense-mediated decay (NMD) [41]. Important to neoantigen discovery, mRNAs carrying nonsense mutation targeted for NMD produce peptides that bind MHC class I for presentation [42], as do mRNAs carrying missense mutations. Thus, these observations raise two important questions: (1) are peptides derived from alternative splicing antigenic? and (2) is there a basis for their use in T cell-based immunotherapy?

Work from Kahles et al. has begun to address these questions. In a pan-cancer analysis across 32 cancer types from The Cancer Genome Atlas and matched normal tissue from The Genotype-Tissues Expression Project, they aimed to identify (1) recurrent alternative splicing events found predominately in cancer tissue, (2) neojunctions–defined as exon-exon junctions found predominately in tumor samples derived from alternative splicing, and (3) alternative splicing-derived putative neoepitopes (ASNs). The latter analysis was carried out in 63 breast and ovarian cancer samples from which they identified at least one SNV-derived putative neoepitope in 19/63 samples and at least one ASN in 43/63 samples, confirmed in the Clinical Proteomic Tumor Analysis Consortium database and predicted to bind MHC class I. Of these putative neoepitopes, 15 ASNs were recurrent within one cancer type, whereas 5 ASNs were found to be recurrent between both cancer types. Conversely, none of the SNV-derived peptides were recurrent between samples [38]. Although Kahles et al. did not go on to interrogate the immunogenicity of the ASNs, several other studies have demonstrated the potential of splicing-derived neoepitopes to elicit an autologous CTL response (Table 2). Otherwise, we are unaware of the successful incorporation of neojunctions into immunotherapies to date. Together, these studies highlight: (1) the prevalence of ASNs and the capability of ASNs to elicit cytotoxic T cell responses and (2) the recurrent nature of this class of cancer antigen among patients, and thus “off-the-shelf” immunotherapies targeting these shared ASNs is worth exploring further.

### 2.4. Identification of Alternative Splicing-Derived Neoantigens

Computational identification of neoantigens derived from alternatively spliced- genes take on a similar, yet modified approach to that of SNV-derived neoantigens. This pipeline begins by aligning RNAseq from tumor samples to normal tissue to identify splicing variants (Table 3). Further comparison to additional healthy tissue databases is highly recommended given the specificity of splicing on tissue and developmental time points and potential immunologic tolerance to self-antigens.

The identification of cancer-specific neojunctions is limited by a variety of factors including a lack of understanding of the level of RNA expression in the cancer cell required to elicit a T cell response; the degree to which variations in mutational loads between cells within a sample will influence their efficacy as a monotherapy or combination therapy; and a consensus on the appropriate matched normal tissue.

The importance of the latter was highlighted by David et al., [55] through analyses of splicing events in cancer and non-cancer RNA-seq data from The Cancer Genome Atlas, the Genotype-Tissue Expression Project, and the Sequence Read Archive. The result of this analysis demonstrated that simply comparing cancer samples to the matched normal tissue from which cancer derives, in that absence of considering other tissue types and developmental time points, results in erroneously identified cancer-specific neojunctions.

**Table 3 pharmaceuticals-14-00993-t003:** Software for identifying alternative splicing events as part of neoantigen discovery.

Server	Access (as of 27 September 2021)	Refs
MISO	http://hollywood.mit.edu/burgelab/miso/	[56]
rMATS *	http://rnaseq-mats.sourceforge.net/rmats3.2.4/	[57]
MAJIQ *	https://majiq.biociphers.org	[58]
LeafCutter	https://github.com/davidaknowles/leafcutter/	[59]
SplAdder *	https://github.com/ratschlab/spladder	[60]
JUM	https://github.com/qqwang-berkeley/JUM	[61]
Whippet	https://github.com/timbitz/Whippet.jl	[62]

* detect novel splicing events in addition to annotated events.

### 2.5. Endogenous Retroviruses

Endogenous retroelements and retroviruses are sequences leftover from germ-line infections of exogenous retroviruses that are stably integrated into the host genome. Many HERV fossils have accumulated mutations and deletions, and, in cancers, are subject to a global loss of DNA methylation. In the human genome most, but not all, HERVs are “extinct” in that they cannot produce infectious viruses [63]. Experiments performed in the 1990s to the current day show that the expression of an endogenous Murine Leukemia Virus is greatly increased in mouse tumors and it presents an immunodominant peptide [64,65]. Sophisticated algorithms that make the analysis of repeated sequences possible have led to a battery of recent studies which show differential expression of HERVs may be used as biomarkers or for increased immunogenicity in immunotherapies, as discussed below.

The long terminal repeat (LTR) sequences of endogenous retroviruses are usually heavily methylated and inactive [66]. Treatments with demethylating agents such as 5-azacitidine (AZA) have pleiotropic effects, including bidirectional transcription from HERV LTRs. When double-stranded RNA is formed, it is recognized by intracellular sensors such as RIG-I and MDA-5 which initiate the interferon pathway resulting in increased antitumor immunity. Specifically, MHC molecules are upregulated, and proliferation is decreased. In addition, proteins or protein remnants from HERVs can be translated and enter the antigen processing and presentation pathways (Figure 3). HERVs that were silenced throughout immune cell development and have a corresponding repertoire of T cells that recognize them as foreign, will stimulate an immune response.

A number of studies are underway to determine the immunogenicity of particular HERVs in cancers. The Childs group has identified an HLA-A2- [67] and HLA-A11- [68] restricted HERV-E-specific T cells in clear cell renal cell carcinomas, the most common form of kidney cancer, and very little expression in normal kidneys. They are currently recruiting for a clinical trial testing HERV-E TCR-transduced autologous T cells in people with metastatic disease NCT03354390. In a study of 34 patients with myeloid malignancies [69], a tumor-type known to have low mutational burdens, Saini and colleagues used tetramer analysis to determine that 17 of the patients had T cells to 29 HERV-derived peptides, particularly to HERVH-5, HERVW-1, and HERVE-3. These T cells were found at lower frequencies in healthy donors. A study evaluating the expression of HERV loci in head and neck squamous cell carcinomas examined 43 paired tumor and tumor-adjacent RNA-seq datasets available from The Cancer Genome Atlas [70]. 1846 HERVs were expressed in tumor samples compared to 1550 HERVs expressed in normal samples. In this study, HERV expression was associated with increased survival.

**Figure 3 pharmaceuticals-14-00993-f003:**
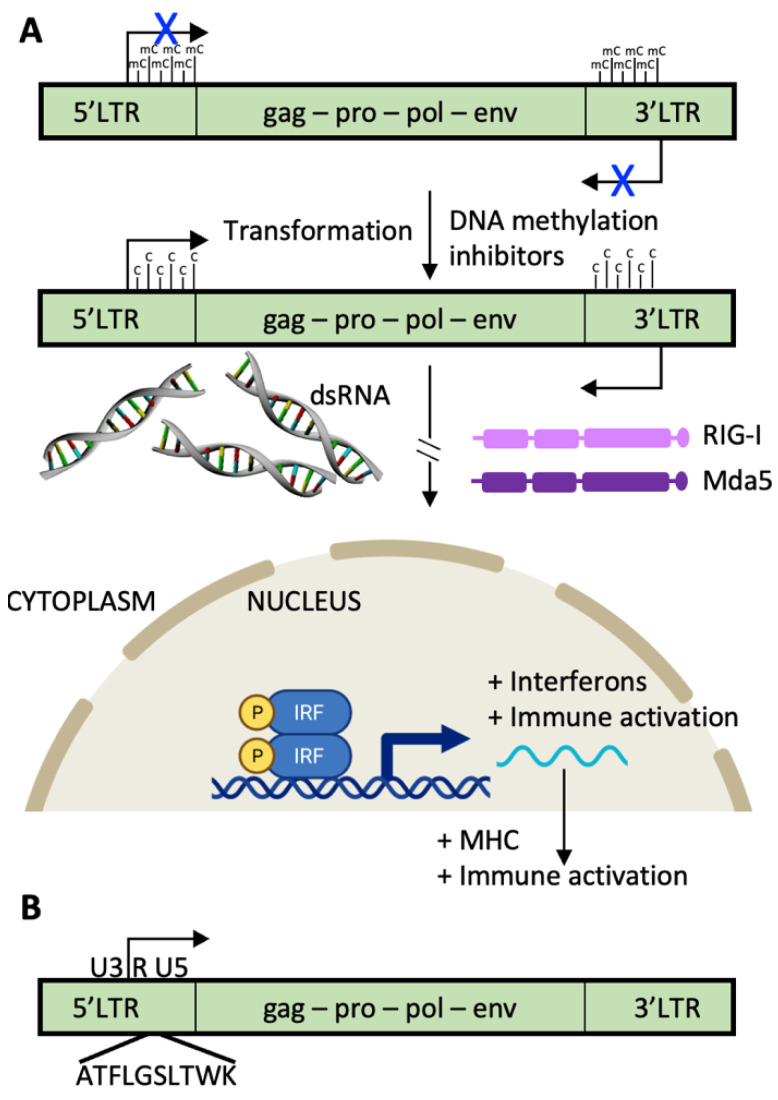
HERVs with productive open reading frames are a source of cellular and humoral tumor antigens. (**A**) The long terminal repeat sequences (LTRs) of most human endogenous retroviruses (HERVs) that express proteins are silenced by epigenetic methylation and demethylated during the transformation process. Since the LTRs transcribe the same sequences in opposite directions, double-stranded HERV RNA molecules are formed, activating cytosolic pattern recognition receptors such as RIG-I and MDA5, leading to activation of the signaling pathways of Type 1 Interferons, and increasing antitumor immunity by upregulation of MHC molecules. (**B**) Example of a HERV tumor antigen expressed in clear cell renal cancers and recognized by cytotoxic T cells is shown. A HERV-E, found on chromosome 6q, harbors the HLA-A11-restricted CT-RCC antigen in the second reading frame downstream of the transcriptional start site in the 5′ LTR (peptide identified in [68]). HERV-E is so-named because the tRNA “primer binding site” of the retrovirus is glutamic acid (single letter code is E). gag, capsid region; pro, protease region; pol, polymerase region; env, envelope region. U3 and U5 are the “Unique” regions of the LTRs containing retroviral enhancers, promoter, transcription start sites, and poly-adenylation sites; R is the “Repeated” region required for proviral DNA synthesis.

Hormone receptor-positive breast cancers show the lowest incidence of lymphocytic infiltration at 6% [71] and low mutational burden [72], both consistent with the low response of patients to checkpoint blockade immunotherapies (reviewed in [73]). Thus, immunogenic opportunities afforded by HERV or neojunctions expression may provide targets. The most investigated HERV in breast cancer is HERV-K. Using data from The Cancer Genome Atlas, Wang-Johanning et al. showed that HERV-K is most highly expressed in basal cell breast cancer [74]. They also showed that serum and mRNA from HERV-K(HML-2) at the time of cancer diagnosis predict metastasis [75]. However, analysis of the remaining regions of the genome has not been investigated thoroughly in breast cancer.

A number of algorithms have been recently developed to map the RNA expression activity of the originally unmapped coding sequences (Table 4). ERVmap provides an annotated map of 124 HERV loci and 3220 proviral ERVs expressed in a cell-type-specific manner [76]. A subset of these was elevated in an IFN-independent signature in peripheral blood of lupus erythematosus patients [77]. A computational tool, Telescope, was used to reassign repetitive unmapped DNA fragments to the most likely transcript of origin and estimate expression at specific HERV loci [78]. This software identified 1365 individual HERV loci that are expressed in one or more cell types. Another tool, REdiscoverTE, was used to analyze repetitive sequences and showed that tumors present not only peptides derived from HERVs, but also other classes of transposable elements like long interspersed nuclear elements (LINE), short interspersed nuclear elements (SINE), and SINE-VNTR-Alu (SVA) [79]. This study considered genomic methylation and found that the expression of 262 transposable elements subfamily resulted from the proximal loss of DNA methylation. Importantly, these subfamilies showed significant overexpression in at least one cancer type from The Cancer Genome Atlas.

### 2.6. TCR T Cell vs. CAR T Cell Response

Current T cell-based immunotherapies either promote an endogenous T cell response (e.g., anti-cancer vaccines or ICI therapy) or deliver an engineered T cell response through adoptive cell therapy (e.g., CAR T cells or transgenic TCR T cells). Through T cell engineering, the specificity of T cells can be redirected by introducing an antigen-specific chimeric antigen receptor (CAR) or transgenic TCR [80]. CAR T cells are distinct from TCR T cells due to a recombinant engineered receptor instead of the physiological TCR (Figure 4). CAR T cells bind their cognate antigen independent of MHC using an extracellular antibody-derived single-chain fragment variable (scFv) domain, fused to intracellular costimulatory domains derived from normal T cell signaling (e.g., CD28, 4-1BB, and CD3) which drive signal activation and amplification within the CAR T cell [81,82,83]. Thus, CAR T cells target surface proteins independent of MHC, whereas TCR T cells target MHC-restricted antigens derived from intracellular or extracellular proteins.

CAR T cells have had remarkable success in treating B cell malignancies and are currently an FDA-approved standard of care for patients with relapsed or refractory B-cell acute lymphoblastic leukemia (B-ALL). However, CAR T cell therapy is not without a number of limitations, including antigen escape, limited efficacy in solid tumors, life-threatening toxicities, limited persistence, and poor trafficking [84]. For example, roughly 10–20% of pediatric patients receiving anti-CD19 CAR T cells relapse due to epitope loss due to deletions, frameshift and missense mutations, and, alternatively, spliced CD19 mRNA [85]. One strategy being employed to address antigen escape is the development of dual or tandem CARs that target multiple antigens. More specifically, the use of anti-CD19/anti-BCMA CAR T cells in multiple myeloma and anti-CD19/anti-CD22 CAR T cells in ALL and diffuse large B-cell lymphoma (DLBCL) have both shown enhanced efficacy versus monotherapy [86,87,88]. Of note, current CAR T cell strategies lack tumor specificity and, in the case of the anti-CD19 CAR, result in B-cell aplasia that must be treated with intravenous immunoglobulin replacement therapy.

Despite the unprecedented success of CAR T cells in hematological malignancies, fewer than 10% of proteins are expressed on the cell surface, rendering the remaining 90% of proteins excluded from being targeted by CAR T cells [89]. Given this limitation, there is a clear advantage of transgenic TCR T cells in the number of target antigens, including neoantigens that confer tumor-specificity. Similar to CAR T cells, TCR T cell therapy is not without unique challenges (Table 5). TCR T cells are HLA-restricted and therefore must be matched with patients’ HLA alleles. An added barrier to TCR T cell therapy is the identification of safe and effective target antigens as well as the TCRs specific to those antigens, which may vary among patients, making this a more personalized form of cancer immunotherapy [90].

A current focus of the field is targeting solid tumors using ACT which has remained a challenge with CAR T and TCR T cells. This is due, in part, to the lack of tumor-specific proteins on solid tumors. This lack of specificity has caused catastrophic clinical outcomes, as seen in patients receiving an anti-HER2 CAR [12]. Severe toxicities have also been observed in transgenic TCR clinical trials, i.e., severe colitis in patients receiving anti-human CEA TCR [92] and severe mental changes, coma, and death in patients receiving anti-MAGE-A3 TCR (later found to be the result of cross-reactivity with MAGE-A12 in the brain) [92]. TCR T cells are currently under investigation against a number of antigens, including TAAs such as NY-ESO-1 (NCT0369137) and MAGE-C2 (NCT04729543), and TSAs, such as HPV E6 (NCT03578406) and KRAS G12V (NCT04146298) [93].

## 3. Summary, Conclusions, and Future Perspectives

Despite the strides made in the identification of putative cancer antigens, and the success of T cell-mediated immunotherapy [94], these therapies provide benefit to a limited subset of patients and cancer types and many of the initial responders do not have long-lasting remission [84]. This has largely been attributed to mechanisms involving immune escape, e.g., antigen escape, tumor heterogeneity and evolution, and restricted trafficking and tumor infiltration of T cells [95,96].

A number of classes of cancer antigens are recognized by autologous T cells and elicit a cytotoxic T cell response. To date, antigens derived from self and tumor-specific antigens have been interrogated as targets of T cell-based immunotherapies. Antigens derived from self are encumbered with immunotolerance and immunosuppression, requiring the use of strong T cell-based adjuvants, co-stimulation, or engineered T cells. Targeting self-antigens in this manner remains a safety concern in terms of on-target, off-tumor effects. Therefore, a focus of the field has been on identifying and targeting tumor-specific antigens.

In multiple cancer types, nsSNVs represent the highest frequency of exonic mutations, followed by neojunction-derived antigens [35,97]. A major limitation in advancing nsSNV-derived neoantigens into clinical targets has been in the identification and translation of peptides that elicit strong CTL responses across patients and cancer types. Nonetheless, there are currently over 100 clinical trials investigating SNV-derived neoantigens across multiple cancer types. Only recently have the tools been developed to explore potential antigens from neojunctions, HERVs, and other repeated elements. Thus, the antigenic potential of these targets is far from realized.

Ultimately, the utility of T cell-based immunotherapies lies in the accurate identification and prediction of which antigens will safely elicit a cytotoxic T cell response in patients. Therefore, optimization of peptide:MHC prediction models that reduce the number of false positives in combination with the development of cytotoxic assays that quickly and accurately measure T cell responses, will facilitate the identification and utilization of neoantigens (no matter the type) in T cell-based immunotherapies.

## Figures and Tables

**Figure 1 pharmaceuticals-14-00993-f001:**
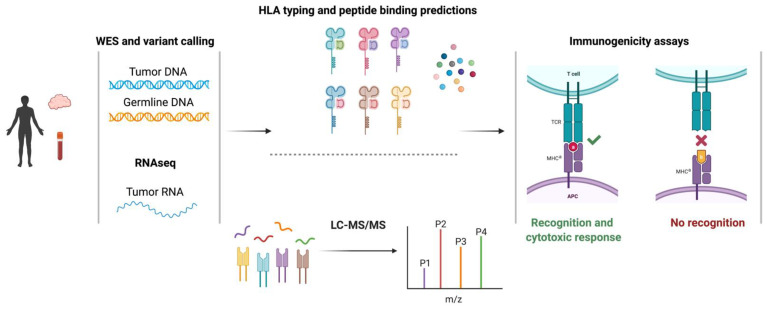
Overview of the pipeline to identify SNV-derived neoantigens. Neoantigens are identified using data from WES, RNA sequencing, and HLA-binding predictions (top), and/or in combination with elution of the antigenic peptides from cancer cells using mass spectrometry (bottom) as described in the text. WES, whole-exome sequencing; LC-MS, liquid chromatography-mass spectrometry.

**Figure 2 pharmaceuticals-14-00993-f002:**
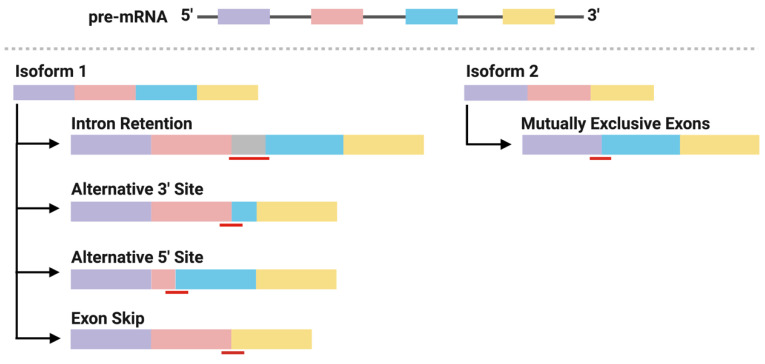
Mutations in cis- and trans-acting splicing elements may form neojunctions predominately expressed in cancer cells through alternative splicing patterns. Schematic view of constitutive splicing of 2 mRNA isoforms from a shared pre-mRNA transcript and their respective alternative splicing patterns forming neojunctions. Black lines represent introns, boxes represent exons and red lines represent neojunctions.

**Figure 4 pharmaceuticals-14-00993-f004:**
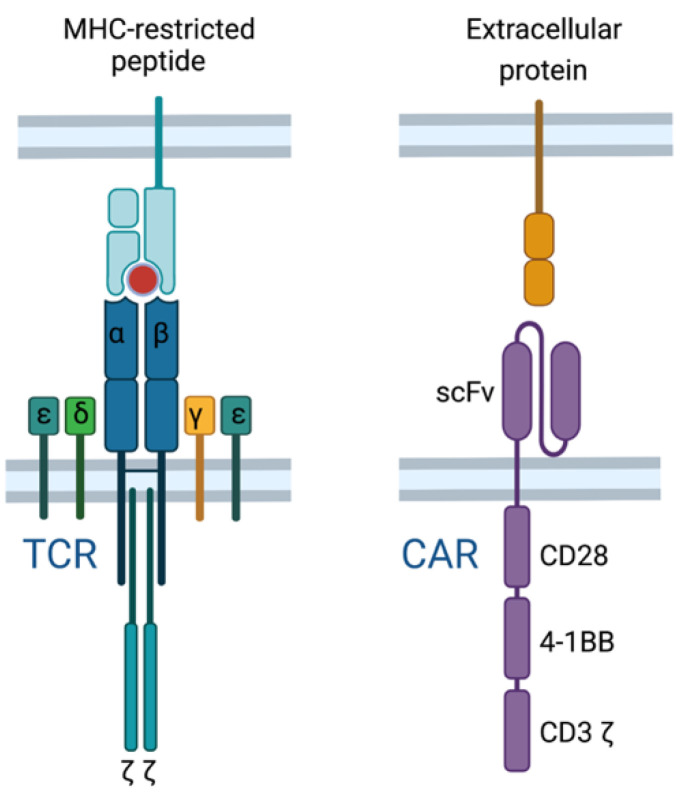
Transgenic T cell receptors (TCRs) and chimeric antigen receptors (CARs) redirect T cell specificity in distinct ways. TCRs recognize MHC-restricted peptides derived from either extracellular or intracellular proteins, whereas CARs bind their cognate MHC-independent antigen through an antibody-derived domain.

**Table 1 pharmaceuticals-14-00993-t001:** MHC binding prediction tools for neoantigen discovery.

Server	Training	Allele	Access (as of 27 September 2021)	Refs
NetMHCpan4.1	BA and EL	MHC I	http://www.cbs.dtu.dk/services/NetMHCpan/	[24]
MHCAttnNet	BA and EL	MHC I & II	https://github.com/gopuvenkat/MHCAttnNet	[25]
MixMHCpred2.1	EL	MHC I	https://github.com/GfellerLab/MixMHCpred	[26,27]
MHCflurry	BA MHCI	MHC I	https://github.com/openvax/mhcflurry	[28]
NetMHCIIpan4.0	BA and EL	MHC II	http://www.cbs.dtu.dk/services/NetMHCIIpan/	[29]
MARIA	BA and EL	MHC II	https://maria.stanford.edu	[30]

Binding affinity, BA; Eluted ligand, EL.

**Table 2 pharmaceuticals-14-00993-t002:** Empirically derived immunogenic HLA class I-restricted peptides derived from alternative splicing.

Gene	Peptide	Tumor	Peptide Origin	Refs
GP100	VYFFLPDHL	Melanoma	Intronic	[43]
MUM1	EEKLIVVLF	Melanoma	Intronic	[44]
AIM2	RSDSGQQARY	Melanoma	Intronic	[45]
TRP-2	EVISCKLIKR	Melanoma	Intronic	[46]
MGAT5	VLPDVFIRC/VLPDVFIRCV	Melanoma	Intronic	[47]
LAGE1	MLMAQEALAFL	Melanoma	aORF	[48]
TRP1	MSLQRQFLR	Melanoma	aORF	[49]
NYESO1	LAAQERRVPR	Breast and melanoma	aORF	[50]
iCE	SPRWWPTCL	Renal cell carcinoma	aORF	[51]
RU2	LPRWPPPQL	Renal cell carcinoma	Intronic	[52]
CD20	RMSSLELVI	Lymphoma	aORF	[53]
Survivin-2B	AYACNTSTL	Oral cancer	aORF	[54]

Alternative open reading frame (aORF).

**Table 4 pharmaceuticals-14-00993-t004:** Software tools for antigen discovery from retroelements.

Server	Access (as of 27 September 2021)	Refs
ERVmap	http://mtokuyama.github.io/ERVmap/	[74]
Telescope	http://github.com/mlbendall/telescope	[75]
REdiscoverTE	https://research-pub.gene.com/REdiscoverTEpaper	[77]

**Table 5 pharmaceuticals-14-00993-t005:** Advantages and Disadvantages of transgenic TCR vs. CAR T cell-based therapy.

	Transgenic TCR T Cells	CAR T Cells
Advantages	○ Recognize peptides derived from intracellular and extracellular proteins ○ High antigen sensitivity (as little as 1 antigen per cell needed) ○ Target antigen may be tumor-specific (neoantigen-derived)	○ Kills cancer cells at a log-scale [91] ○ MHC-independent ○ Not dependent on mutations and neoantigen expression
Disadvantages	○ Receptor is MHC-restricted, necessitating MHC-matching ○ Identifying target antigen and patient-specific TCR remains challenging ○ Severe toxicities	○ Only targets extracellular proteins ○ Severe toxicities, including cytokine release syndrome ○ Lower antigen sensitivity (>100 antigens per cell needed, in general)

## Data Availability

Data sharing not applicable.

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
