# Peer review of "Implications of Antigen Selection on T Cell-Based Immunotherapy"

_pharmaceuticals, 2021, doi:10.3390/ph14100993_

Round 1

Reviewer 1 Report

The manuscript „Implications of Antigen Selection on T cell-based Immunotherapy“  by Faye A Camp, Jill E Slansky is a profound contribution to the scientific discussion on immunotherapy in cancer treatment. It is focused on the selection work which has to be done to reveal the molecular structures most efficiently to target.  The paper is readable also for researchers who are not intimate experts in that field and offers to get a deeper understanding.

I have a few comments

Line-28                        Is dark genome a scientific term?

2.1. Cancer Antigens  Why didn’t you mention carbohydrates as cancer-associated molecular structures? Is the impact low in T cell activation?

Line 58-69                   TAA and esp. HER2/neu are expressed in normal tissue under  conditions for reproduction and inflammation?

Line 112 ff.                  Authors should give some explanations on the interaction between  neoantigens and HLA molecules that impact binding affinity.

Line 115                      Prediction tools are also trained on data from MS. How are peptides  generated? How is the impact of the source of peptides on the  prediction quality (in-gel-digestion vs. LC-MS coupling)?

Line135-140                The sentences here are to long to get the message. Should be rewritten.

Line 155                      Is this a single experiment? Should give more descriptive statitics (e.g. range of SNVs). Which types of genes are detected? Are they in cancer  pathways, e.g. tyrosine kinases, ras, PI3K?                           

Line 162                      Show descriptive statitics (mean, SD, p values)

Line 173                      Chapter number is wrong

Line 279                      Give references

Line 330                      List the software tolls in a table like tab. 3

Author Response

Point 1: Line-28 Is dark genome a scientific term? 

Response 1: Although becoming a popular term, “dark genome” does sound jargony, so we have changed it to retroelements and HERVs throughout the document.

Point 2: 2.1. Cancer Antigens Why didn’t you mention carbohydrates as cancer-associated molecular structures? Is the impact low in T cell activation?

Response 2: Thank you for identifying this oversite. We added tumor-associated carbohydrate antigens (TACAs) to the list of tumor-associated antigens (TAAs) (line 66).  Initially left out due to knowing TACAs alone evoke a poor T-cell-dependent immune response.  There is also a high burden of immunotolerance and immunosuppression using naturally occurring TACAs with T cell-based immunotherapies.

Point 3: Line 58-69 TAA and esp. HER2/neu are expressed in normal tissue under conditions for reproduction and inflammation? 

Response 3: HER2 is amplified in 15-30% of breast cancers, dramatically upregulating the surface expression of this receptor.  We added more detail about the overexpression of HER2 in breast cancer to clarify this point (line 64-66).

Point 4: Line 112 ff. Authors should give some explanations on the interaction between neoantigens and HLA molecules that impact binding affinity. 

Response 4: Agree, this is now addressed in lines 124-131 (track changes document).

Point 5: Line 115 Prediction tools are also trained on data from MS. How are peptides generated? How is the impact of the source of peptides on the prediction quality (in-gel-digestion vs. LC-MS coupling)?

Response 5: Agree, this is now addressed in lines 122-136 (track changes document).

Point 6: Line135-140 The sentences here are too long to get the message. Should be rewritten. 

Response 6: Thank you, we separated sentences to make them easier to understand. (line 153-159)

Point 7: Line 155 Is this a single experiment? Should give more descriptive statistics (e.g. range of SNVs). Which types of genes are detected? Are they in cancer pathways, e.g. tyrosine kinases, ras, PI3K? 

Response 7: In this paper, the authors analysed 31 patient samples for exonic mutations.  The following details have been added to the text to clarify:  nsSNVs represented 62.4% of all exonic mutations; the top 3 genes containing nsSNVs; genes that contained nsSNVs that were recurrent between at least 2 patient samples. (now lines 175-180).

Point 8: Line 162 Show descriptive statistics (mean, SD, p values) –

Response 8: We added statistics to data including n, median, range and p value (now lines 180-188).

Point 9: Line 173 Chapter number is wrong.

Response 9: Thank you for identifying this oversite.  This and subsequent chapter numbers have been updated.

Point 10: Line 279 Give references.

Response 10: Since this is the topic of the section, we added “as discussed below.”

Point 11: Line 330 List the software tools in a table like tab. 3. 

Response 11: A new table, similar to table 3, has been added to the end of the HERV section (line 344).

Reviewer 2 Report

The review article presented by Faye A. Camp and Jill E. Slansky has the purpose to shed light on some important issues related to T cell-based immunotherapies. Authors clearly summarize and discuss the various types of cancer antigens, focusing on antigens from neojunctions and the dark genome, and the implications they have on the outcome of T cell-based immunotherapy.

Tables and figures are clear and help the reader to better follow the text.

The review is interesting and includes a balanced, comprehensive and critical view of the research area. It is well written and easy to read

Minor points:

- Authors should carefully review references section:

  1. DOI Number is missing in Ref: 9, 29, 34, 53, 58, 73
  2. Ref. 13 number of pages is missing
  3. Ref. 62, 64, 84: the name of the Journal and the number of volume should be written in italics

Author Response

Point 1: Tables and figures are clear and help the reader to better follow the text.  The review is interesting and includes a balanced, comprehensive and critical view of the research area. It is well written and easy to read

Response 1: Thank you for your positive comments!

Point 2: - Authors should carefully review references section:

  1. DOI Number is missing in Ref: 9, 29, 34, 53, 58, 73
  2. Ref. 13 number of pages is missing
  3. Ref. 62, 64, 84: the name of the Journal and the number of volume should be written in italics

Response 2: All references have been re-formatted according to the MDPI EndNote style. This corrected all of the errors listed in point 2.

Reviewer 3 Report

The authors reviewed the current knowledge of cancer antigens, how to identify cancer neoantigens, and their use in combination with T-cell-based immunotherapy. The content of this review is comprehensive and easy to read. I have a couple of comments like below

Major

  • Structurally, the manuscript consists of (1) Introduction and (2) T-cell mediated therapy review. Please add a summary, conclusions, and future perspectives at the end of the manuscript. The limitation of this research should also be included in the review. What is the bottleneck of the current research?
  • While the presence of neoantigens derived from alternative splicing is well described, I could not confirm that these types of neoantigens are targetable for the treatment. Is there any pre-clinical or clinical data showing this point?

Minor

  • As is reviewed there is three class of cancer antigens known to relate with T cell immunity. Is it known which of the class is dominant in general? I think I never read reviews mentioning this.
  • From the clinical aspect of view, the correlation of the tumor type and tumor antigen is an important issue. Please add which type of antigen is more common in which types of cancer.
  • Section 2.1. Cancer Antigens

Please add the percentage of identifying the cancer antigens among cancer patients. How often these antigens cannot be identified.

  • Page 8 line 334. Please add a description about “telescope” before quoting.
  • Like the authors mentioned, a limited number of patients respond to the current T cell-mediated therapy. Please include some of the most recent clinical trial data of adoptive T cell transfer to show what the current reality is.

Author Response

Point 1: Structurally, the manuscript consists of (1) Introduction and (2) T-cell mediated therapy review. Please add a summary, conclusions, and future perspectives at the end of the manuscript. The limitation of this research should also be included in the review. What is the bottleneck of the current research?

Response 1: Thank you. We have added a third section: “3. Summary, conclusions and future perspectives” that encompasses the main points of the review (line 404).

Point 2: While the presence of neoantigens derived from alternative splicing is well described, I could not confirm that these types of neoantigens are targetable for the treatment. Is there any pre-clinical or clinical data showing this point?

Response 2: To date, we are unaware of successful incorporation of neojunctions into immunotherapies.  Discussed further in lines 233-236 and table 2.

Point 3: As is reviewed there is three class of cancer antigens known to relate with T cell immunity. Is it known which of the class is dominant in general? I think I never read reviews mentioning this.

Response 3: The dominant class of cancer antigen is nsSNV-derived although burden is highly variable between patients and cancers types. 

Point 4: From the clinical aspect of view, the correlation of the tumor type and tumor antigen is an important issue. Please add which type of antigen is more common in which types of cancer.

Response 4:  Stats for breast cancer were added (line 175).

Point 5:     Section 2.1: Please add the percentage of identifying the cancer antigens among cancer patients. How often these antigens cannot be identified.

Response 5: I added the following stats: NY-ESO-1 can be found in up to 40% of breast cancers; MART-1 is expressed in >90% of melanoma; KRAS mutants account for 60–70% of pancreatic cancers and 20–30% of colorectal cancers. (lines 61-65; 81-84)

Point 6: Page 8 line 334. Please add a description about “telescope” before quoting.

Response 6: Thank you, we have fixed the working for clarity.

Point 7: Like the authors mentioned, a limited number of patients respond to the current T cell-mediated therapy. Please include some of the most recent clinical trial data of adoptive T cell transfer to show what the current reality is.

Response 7:   Thank you. We added more information specific to CAR T cells and transgenic T cells (lines 365-403).
